# Order-disorder phase transition driven by interlayer sliding in lead iodides

Seyeong Cha [1,4], Giyeok Lee [2,4], Sol Lee[1], Sae Hee Ryu[3], Yeongsup Sohn[1], Gijeong An[1], Changmo Kang[1], Minsu Kim[1], Kwanpyo Kim [1], Aloysius Soon [2] ✉ & Keun Su Kim [1] ✉

A variety of phase transitions have been found in two-dimensional layered materials, but some of their atomic-scale mechanisms are hard to clearly understand. Here, we report the discovery of a phase transition whose mechanism is identified as interlayer sliding in lead iodides, a layered material widely used to synthesize lead halide perovskites. The low-temperature crystal structure of lead iodides is found not 2H polytype as known before, but non-centrosymmetric 4H polytype. This undergoes the order-disorder phase transition characterized by the abrupt spectral broadening of valence bands, taken by angle-resolved photoemission, at the critical temperature of 120 K. It is accompanied by drastic changes in simultaneously taken photocurrent and photoluminescence. The transmission electron microscopy is used to reveal that lead iodide layers stacked in the form of 4H polytype at low temperatures irregularly slide over each other above 120 K, which can be explained by the low energy barrier of only 10.6 meV/atom estimated by first principles calculations. Our findings suggest that interlayer sliding is a key mechanism of the phase transitions in layered materials, which can significantly affect optoelectronic and optical characteristics.

Two-dimensional (2D) layered materials have continued to attract broad interest owing to their potential for application in nanoelectronics and optoelectronics[1–3]. They also serve as a platform to study various fundamental quantum phenomena, such as phase transitions. Indeed, this class of materials has been widely used to explore various phase transitions that include charge density waves[4–11], magnetism[12–14], and (order-order) structural transitions[13–16]. Recently, it has been increasingly recognized that the typically weak interlayer coupling and stacking shifts in these materials play a key role in phase transitions[9–11]. However, it has been challenging to clearly understand their atomic-scale mechanism and, more crucially, how it is linked to the material's optoelectronic and optical characteristics.

Lead iodide (PbI$_2$) is a layered material, the powder form of which has been widely used to synthesize lead halide perovskites[17,18]. In the single layer of PbI$_2$, there is the 2D hexagonal network of Pb atoms, sandwiched by those of I atoms as in the typical form of 1 T transition metal dichalcogenides (TMDs) (Fig. 1a). This is the unit layer stacked on top of each other by a weak van der Waals force, which is the 2H polytype in the conventional nomenclature of PbI$_2$ (Fig. 1b)[19]. This nomenclature is different from that of TMDs, and to prevent confusion, we will hereafter use the nomenclature of PbI$_2$. The 2H phase has been widely accepted as the most common stacking structure of bulk PbI$_2$[19–30], although some of the recent Raman and X-ray diffraction (XRD) studies reported puzzling results indicative of other polytypes[31–33]. In the corresponding band structure of 2H-PbI$_2$, there are the conduction band minimum and the valence band maximum at the A point with the energy gap ($E_g$) of ~2.4 eV (Fig. 1e)[26–28]. This directness of $E_g$ whose magnitude corresponds to the spectral range of visible light renders PbI$_2$ as a promising material for application in optoelectronics[30–36], for example,

[1]Department of Physics, College of Science, Yonsei University, Seoul, Korea. [2]Department of Materials Science and Engineering, Yonsei University, Seoul, Korea. [3]Advanced Light Source, E. O. Lawrence Berkeley National Laboratory, Berkeley, CA, USA. [4]These authors contributed equally: Seyeong Cha, Giyeok Lee. ✉e-mail: aloysius.soon@yonsei.ac.kr; keunsukim@yonsei.ac.kr

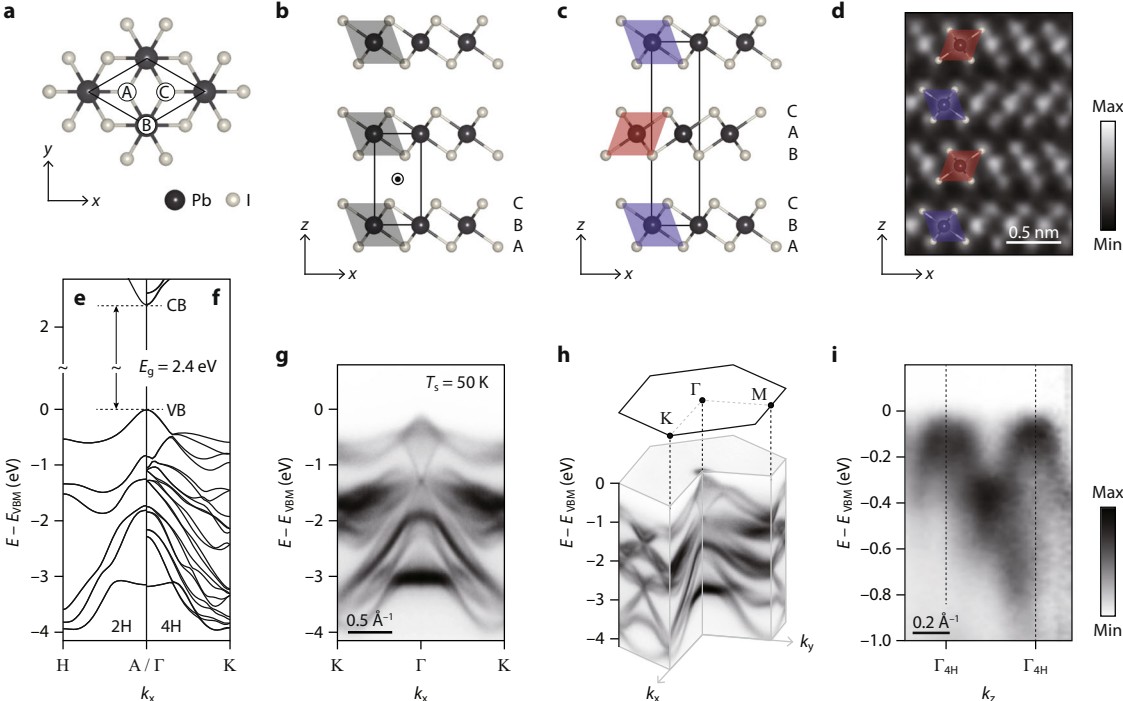

**Fig. 1 | Low-temperature crystal and band structures of bulk PbI₂. a** Ball-and-stick model of single-layer PbI₂ shown from the top view. The black rhombus is the unit cell, in which the 3 high-symmetry sites are marked by A, B, and C. **b, c** Stacking structure of (**b**) 2H-PbI₂ and (**c**) 4H-PbI₂ in the conventional nomenclature of PbI₂ (note that they correspond to 1T and 2H in the nomenclature of TMDs). Black rectangles in **b, c** are their respective unit cells, and the circled dot in **b** marks the inversion centre. The label A, B, and C in **b, c** are the index of atomic sites as defined in **a. d** HAADF-STEM image of PbI₂ taken along the $y$ axis. Red and blue rhombuses in **c, d** represent the octahedron unit of PbI₆ alternatively tilted in opposite directions in different layers, while those tiled in the same directions are shown in grey in **b. e, f** Theoretical band structure of (**e**) 2H-PbI₂ and (**f**) 4H-PbI₂, calculated by DFT considering spin-orbit coupling (see Supplementary Fig. 1 for the fuller set of DFT calculations). CB and VB are conduction and valence bands, respectively. **g** Experimental band structure of bulk PbI₂, taken by ARPES along $k_x$ with the photon energy of 96 eV (corresponding to $\Gamma_{4H}$ in $k_z$). **h** ARPES data taken over $k_x$ and $k_y$ and displayed along a couple of high-symmetry directions. **i** $k_z$ dispersion taken by the photon-energy dependence of ARPES data. Dotted lines show the Γ points of 4H-PbI₂. Source data are provided as a Source Data file.

the photodetector with a low dark current owing to its high resistivity over $10^{13}\ \Omega\cdot\mathrm{cm}$[35,36].

In this work, we first clarify that the low-temperature structure of PbI₂ is not the 2H polytype as known before, but 4H polytype shown in Fig. 1c. More importantly, whereas it has been well known that most lead halide perovskites show two-step phase transitions as a function of temperature[37], no such phase transition was reported in its 2D counterpart, PbI₂. Here, we report the discovery of order-disorder phase transitions characterized by an abrupt spectral broadening of valence bands taken by angle-resolved photoemission spectroscopy (ARPES) at the critical temperature ($T_c$) of 120 K. This transition is accompanied with drastic changes in simultaneously taken photo-conductivity and photoluminescence. Transmission electron micro-scopy (TEM) is employed to reveal that PbI₂ layers stacked in the form of 4H polytype at low temperatures irregularly slide over each other as the sample temperature exceeds 120 K. This can be naturally explained by the low energy barrier of only 10.6 meV/atom estimated by density-functional-theory (DFT) calculations. These findings suggest that interlayer sliding is a key mechanism of the order-disorder phase transitions in a layered material consisting of 2D van der Waals crystals owing to their weak interlayer coupling.

## Results

### Low-temperature structure of PbI₂

Let us first clarify the crystal structure of bulk PbI₂ at low temperatures. For the surface unit cell of PbI₂ shown in Fig. 1a, one can define 3 high-symmetry sites indexed by A, B, and C. The single layer of PbI₂ consists of a 2D hexagonal network of Pb atoms in B sites and those of I atoms in A and C sites, which forms tilted PbI₆ octahedrons shown by the grey

rhombus in Fig. 1b. For this 2H phase in the nomenclature of PbI₂, identical octahedrons are stacked right on top of each other, as in the well-known stacking of 1T-TMDs[4] in the nomenclature of TMDs. This is the hitherto widely accepted as the most common stacking of bulk PbI₂[19–30].

On the other hand, scanning TEM (STEM) data taken from PbI₂ samples indicate otherwise. As shown in Fig. 1d, there are large and small bright protrusions, which represent Pb and I atoms, respectively. This result is clearly against the 2H polytype phase in that Pb atoms in the adjacent layers are not on top of each other. More importantly, the tilted octahedrons, which are expected for 2H-PbI₂ to be identical in different layers, are found not identical, but alternating with their mirror-symmetric pair shown by red and blue rhombuses overlaid in Fig. 1d. In fact, STEM results suggest another polymorph of PbI₂, the so-called 4H polytype illustrated in Fig. 1c. This mirror-symmetric pair of tilted octahedrons are stacked in the sequence of A-B-C and B-A-C, thereby doubling periodicity in the out-of-plane direction ($z$). Another key difference of 4H-PbI₂ from 2H-PbI₂ is that the inversion centre indicated by the circled dot in Fig. 1c is absent in Fig. 1d, namely, 4H-PbI₂ is noncentrosymmetric.

This 4H-polytype structure of bulk PbI₂ at low temperatures can be further supported by ARPES measurements. Figure 1e shows the theoretical band structure of 2H-PbI₂ along the AH direction, where the valence band maximum (VBM) is located (Supplementary Fig. 1), which is overall consistent with previous reports[26,27]. In the energy range from the VBM to −4 eV, there are 6 valence bands of Pb 6$s$ and I 5$p$ char-acters. This number of valence bands should be doubled in 4H-PbI₂ by band folding in $k_z$, given that the unit cell is doubled in $z$ and that the band crossing is protected by 6-fold screw rotation symmetry (Fig. 1d).

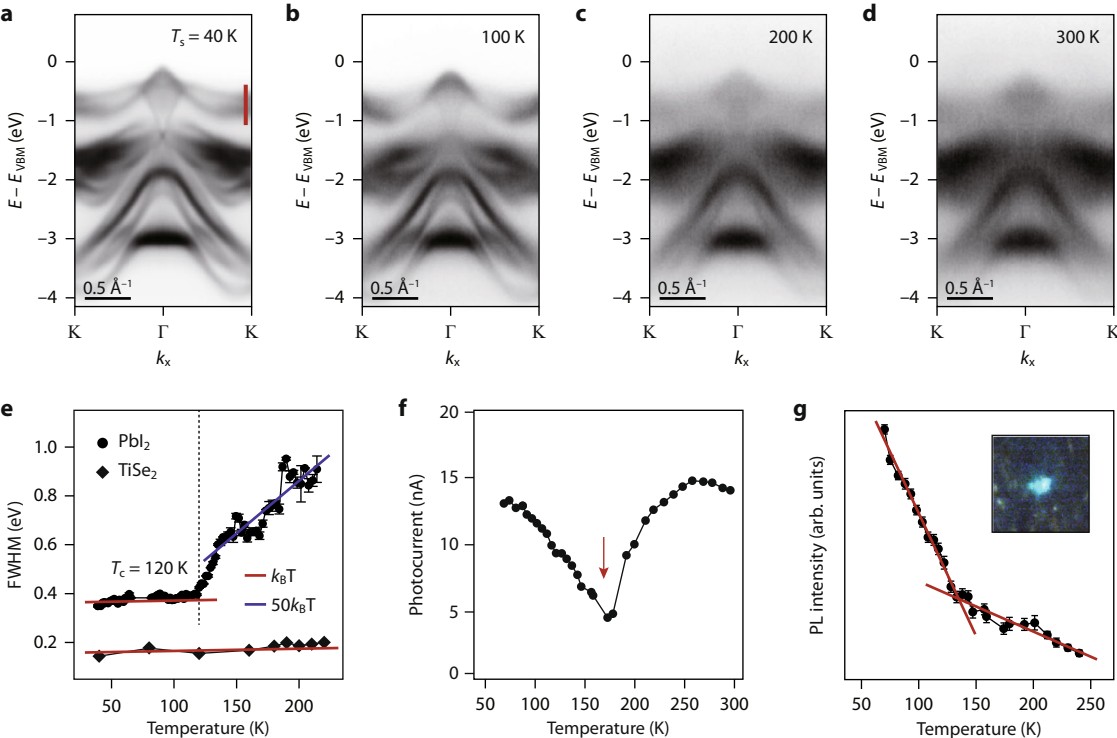

**Fig. 2 | Order-disorder phase transition and its effect on material properties.**
**a–d** Series of ARPES data taken at $T_s$ marked at the upper right of each panel. These data were taken in $k_x$ with the photon energy of 96 eV. **e** FWHM plotted as a function of $T_s$, obtained by a curve fit to energy-distribution curves taken along the red line in **a**. Red and blue lines overlaid are the rate of increase in FWHM expected for $k_B T$ and 50 $k_B T$, respectively. The dotted line shows the $T_c$ of 120 K. The same analysis applied to our ARPES spectra taken from 1T-TiSe$_2$, where the order-order

phase transition is reported at $T_c = 200$ K[4], is given as a reference. **f** Photocurrent taken in the process of ARPES together with ARPES data in **a–d** and plotted as a function of $T_s$. The red arrow indicates the dip in photocurrent. **g** PL intensity of green light emission (inset) taken in the process of ARPES with the excitation energy of 96 eV. The red lines overlaid show two distinct slopes of PL intensity versus $T_s$. The error bars in **e**, **g** are the standard deviation from least-squares fitting. Source data are provided as a Source Data file.

Furthermore, the effect of sizable spin-orbit coupling and broken inversion symmetry in 4H-PbI$_2$ leads to the Rashba-type spin splitting[38], which once again doubles the number of valence bands. Therefore, the total number of valence bands expected for 4H-PbI$_2$ is as many as 24 (Fig. 1f). The experimental band structure measured by ARPES is present in Fig. 1g, h, which clearly favours 4H-PbI$_2$ in terms of the number of valence bands. On top of that, the $k_z$ dispersion of lowest-energy valence bands in Fig. 1i exhibits the periodicity of 0.45 Å$^{-1}$ corresponding to that of 4H-PbI$_2$. From these results, we conclude that the low-temperature structure of bulk PbI$_2$ is not the centrosymmetric 2H polytype, but the noncentrosymmetric 4H polytype. This low-temperature structure can also be supported by non-destructive XRD experiments (Supplementary Fig. 2).

**Phase transition and material properties**
Let us now turn our attention to the observation of a phase transition in this 4H phase with increasing the sample temperature ($T_s$). Figure 2a–d shows a series of ARPES data taken with the same experimental conditions but $T_s$ from 40 K to 300 K. Little is changed between 40 K and 100 K except for the typical thermal broadening effects. At $T_s = 200$ K (Fig. 2c), however, we found the significant broadening of ARPES spectra, which persists up to 300 K (Fig. 2d). For the quantitative analysis, the full width at half maximum (FWHM) of energy-distribution curves taken along the red line in Fig. 2a is plotted as a function of $T_s$ in Fig. 2e. In the range of 40–120 K, the increase in FWHM is as expected for electron-phonon coupling, in the order of $k_B T$ (red line overlaid)[39], where $k_B$ is the Boltzmann constant. In the $T_s$ range greater than 120 K, however, we found the highly abrupt increase in FWHM with the energy scale of 50 times $k_B T$ (blue line overlaid), which cannot be explained by electron-phonon coupling.

This degree of spectral broadening can be accounted for by the abrupt increase of disorder with $T_s$, that is, the order-disorder-type phase transition. Note that this is clearly different from the order-order phase transition in 1T-TiSe$_2$ (Fig. 2e), in which the increase of FWHM follows roughly $k_B T$ across $T_c$. From a curve fit to the inverse FWHM versus $T_s$[40], we estimate $T_c$ to be 120 K (dotted line in Fig. 2e). This order-disorder phase transition is found fully reversible and unexceptionally observed in all of our samples (see Supplementary Fig. 3 for the fuller set of ARPES data).

To see if spectral broadening in electronic structures affects optoelectronic characteristics, the photocurrent was simultaneously measured with ARPES data in Fig. 2a–d and plotted as a function of $T_s$ in Fig. 2f. In the $T_s$ range up to 160 K, photocurrent monotonically decreases with increasing $T_s$, but it turns into the increase near 160 K, forming a dip indicated by the red arrow. The similar anomaly near $T_c$ has been observed in other phase transitions[41], which has been attributed to either a crossover in the type of carriers or divergent scattering in the vicinity of $T_c$ due to the coexistence of competing phases and their phase fluctuations[42]. For the latter, the effect of inhomogeneity near $T_c$ makes charge carriers strongly localized, leading to a dip in conductivity or a peak in resistivity. Indeed, this peak in resistivity near $T_c$ is also manifested by charging effects in Pb 5$d$ core-level spectra (Supplementary Fig. 4).

On the other hand, the sign of this phase transition can also be seen in optical characteristics. In Fig. 2g, we show a plot of photo-luminescence (PL) intensity as a function of $T_s$, taken with the syn-chrotron radiation of 96 eV simultaneously with ARPES spectra and photocurrent in Fig. 2a–f. At low temperature (70 K), one can clearly see green light emission (inset), which is a natural consequence of direct $E_g$ whose magnitude is 2.4 eV. The intensity of green light

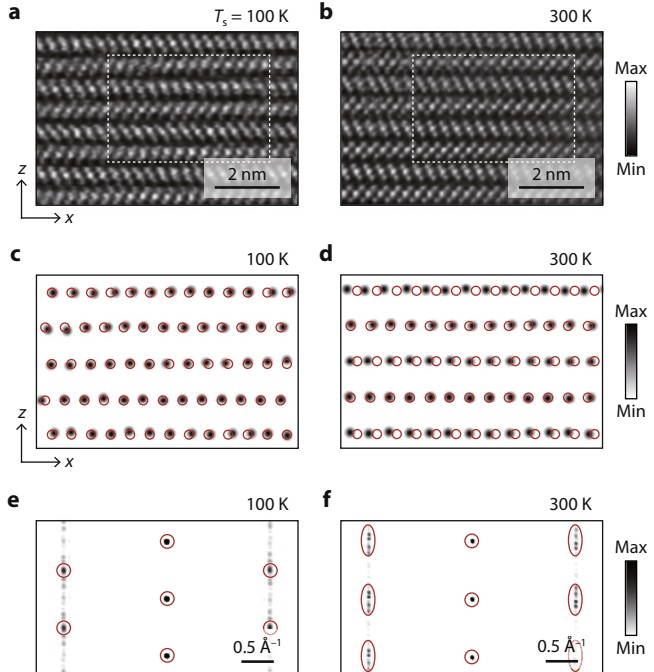

**Fig. 3 | Atomic-resolution imaging above and below $T_c$. a, b** HAADF-STEM images of $PbI_2$ at (**a**) 100 K and (**b**) 300 K taken along the $y$ axis. **c, d** Distribution of Pb atoms taken by the curve fit to large bright protrusions in the dashed box of **a, b** with the 2D Gaussian function. Red open circles mark the location of Pb atoms expected for 4H-$PbI_2$. **e, f** Fourier-transform images of those in **c, d** over the area of $10 \times 10$ nm$^2$. Red circles in **e** indicate the major reciprocal lattice peaks, and red ovals in **f** indicate those spread along the $z$ axis. Source data are provided as a Source Data file.

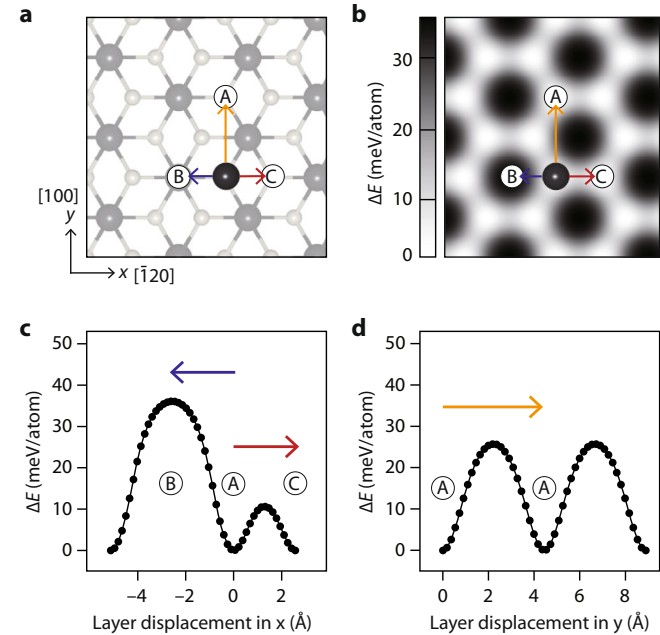

**Fig. 4 | Energy barrier of interlayer sliding between bilayer $PbI_2$. a** Ball-and-stick model of bilayer $PbI_2$ shown from the top view. For clarity, the lower layer is shaded, and only one Pb atom in the upper layer (black ball) is shown. **b** Calculated energy difference of displacing the upper layer with respect to the lower layer and plotted over the same area as in **a**. The circled A, B, and C show the high-symmetry sites in the lower layer, as defined in Fig. 1a. Red, blue, and yellow arrows indicate 3 representative sliding directions as introduced in **a**. **c, d** Energy difference plotted as a function of the layer displacement (**c**) along blue and red arrows, and (**d**) along yellow arrow. Source data are provided as a Source Data file.

decreases with increasing $T_s$, and its slope abruptly varies at around 130 K (red lines overlaid). The similar anomalous temperature dependence of PL associated with the order-disorder phase transition was found in other systems[43]. In the disorder phase, the energy and crystal momenta of conduction and valence bands (and so the bandgap) are ill-defined with the considerable uncertainty of 50 $k_BT$. Then, the scattering of electrons and holes is dominated by various sources of disorder potentials rather than radiative electron-hole recombinations. In this sense, the observed change in the slope of PL seems to suggest a crossover between the two regimes, where the scattering of electrons and holes is dominated by either intrinsic (phonon and electron-hole recombinations) or extrinsic (disorder) factors.

**Mechanism of the phase transition**

To elucidate the mechanism of this phase transition, we employed high-resolution STEM measurements. Figure 3a, b shows high-angle annular dark-field (HAADF) STEM data taken at 100 K and 300 K along the $y$ axis. It is unexceptionally observed for both $T_s$'s that the mirror-symmetric pair of the tilted $PbI_6$ octahedrons (shown by red and blue rhombuses in Fig. 1c) alternates with each other in different layers. On the other hand, the spatial distribution of Pb atoms is selectively taken by a curve fit to large bright protrusions in the dashed box of Fig. 3a, b with 2D Gaussian function, and is compared between 100 K and 300 K in Fig. 3c, d. It can be readily identified for $T_s = 100$ K (Fig. 3c) that most Pb atoms are found in or at least close to red open circles that mark the expected location of Pb atoms for 4H-$PbI_2$. However, in $T_s = 300$ K (Fig. 3d), we found that those in several layers are laterally shifted from 4H-$PbI_2$.

The stacking correlation over the wider region is examined on average by taking the Fourier transform of STEM images, as compared in Fig. 3e, f. Whereas the reciprocal-lattice peaks at 100 K (red circles, Fig. 3e) are overall in good agreement with the pattern for the 4H polytype, those at 300 K (red ovals, Fig. 3f) show a spread of peaks in the $z$ axis. The diffraction pattern taken over the wide area of $80 \times 80$ nm$^2$ in Supplementary Fig. 5 exhibits the clearer streak, indicative of disorder (or the randomness of interlayer sliding) in the $x$ axis. On the contrary, no such interlayer sliding disorder is observed in STEM data taken in the $y$ direction for the smaller area of $3 \times 4.5$ nm$^2$, where one of the three equivalent sliding directions is dominant (Supplementary Fig. 6). This result not only rules out the possibility of electron-beam effects, but also indicates the strong in-plane anisotropy of interlayer sliding.

The in-plane anisotropy can be explained by the energy barrier of interlayer sliding between $PbI_2$ layers. We have performed DFT calculations based on a simple model of bilayer 4H-$PbI_2$ illustrated in Fig. 4a. Figure 4b shows the energy difference $\Delta E$ of displacing the upper layer with respect to the lower layer over the same area as in Fig. 4a. From this $\Delta E$ map, one can clearly see that the energy barrier of interlayer sliding exhibits 3-fold anisotropy. To make the more quantitative analysis, the $\Delta E$ of interlayer sliding is compared in Fig. 4c, d between $x$ and $y$ directions, more generally, between $\langle 110 \rangle$ and $\langle 100 \rangle$ directions. For the case of interlayer sliding along the $\langle 110 \rangle$ directions, Pb atoms in the upper layer can be displaced towards either Pb or I atoms in the lower layer (from A sites to B or C sites in Fig. 4a, b), as indicated by blue and red arrows, respectively. The energy barrier of interlayer sliding along the red arrow (from A to C) is about 10.6 meV/atom that is interestingly comparable to $T_c$. It is, however, about 2.4–3.4 times lower than those along blue and yellow arrows, which explains 3-fold in-plane anisotropy of interlayer sliding. Note that this sliding easy direction (red arrows) corresponds to the transition path along which the 4H stacking configuration of $PbI_2$ returns to its original state at the least energy cost.

## Discussion

We would clarify the difference between stacking disorder and sliding disorder discussed in the present study. As in the case of ice, diamond, silicon carbides[44–46], stacking disorder refers to the aperiodic arrangement of a few well-defined stacking structures, for example, the cubic and the hexagonal. Unlike this stacking disorder, the displacement of a layer with respect to adjacent layers can be anywhere in between A and C sites (red arrow in Fig. 4c) once the energy barrier can be overcome at given $T_s$. Namely, there exists a 2D network of low-energy paths (nearly white regions in Fig. 4b) along which the layers are nearly free to slide over to each other, as in the melting transition[40]. In a broad sense, it may be viewed as a kind of stacking disorder, but their atomic-scale mechanisms are different. The mechanism found here for the phase transition of 4H-PbI$_2$ is a natural consequence of weak interlayer coupling generic to a layered material of 2D van der Waals crystals. Recently, the potential importance of stacking disorder has been increasingly recognized in the study of 1T-TaS$_2$[9,10], but relatively less attention was paid to the possibility of interlayer sliding. In line with that, our results suggest that interlayer sliding is a key mechanism of the order-disorder phase transitions that can significantly affect optoelectronic and optical characteristics.

In summary, we reveal the order-disorder phase transition driven by interlayer sliding in PbI$_2$ with ARPES, TEM, and DFT. The low-temperature crystal structure of bulk PbI$_2$ is identified as the non-centrosymmetric 4H polytype. It undergoes the order-disorder phase transition characterized by the abrupt spectral broadening of valence bands above 120 K with the energy scale about 50 times greater than $k_B T$. It is also accompanied by drastic changes in simultaneously taken photoconductivity and photoluminescence. The PbI$_2$ layers stacked in the form of 4H polytype at low temperatures is found to irregularly slide over each other not in the $\langle 100 \rangle$ directions, but in the $\langle 110 \rangle$ directions. This interlayer sliding with strong in-plane anisotropy can be explained by the low energy barrier of only 10.6 meV/atom in only one direction of $\langle 110 \rangle$ with 3-fold symmetry. Our results suggest that interlayer sliding is a key mechanism of the phase transitions in the layered material of 2D van der Waals crystals.

## Methods

### ARPES experiments

ARPES measurements were carried out at Beamline 7.0.2 (MAESTRO), Advanced Light Source (ALS). This end-station is equipped with the hemispherical analyser whose instrumental energy and angular resolutions are better than 10 meV and 0.1°. The high-flux synchrotron radiation with the photon energy of 60–250 eV was focused on the surface of samples for the beam size of $50 \times 50$ μm$^2$. The single-crystal bulk PbI$_2$ samples (99.9999%, 2D Semiconductors) were glued on the sample holders using conductive silver epoxy. The clean surface of PbI$_2$ was in situ prepared for ARPES measurements by cleaving the samples using the wobble stick in the ultrahigh vacuum chamber whose base pressure was better than $4 \times 10^{-11}$ torr. The kinetic energy of ARPES data was converted to the energy relative to VBM due to charging effects. The emission angle of ARPES data was converted to $k$ based on the momentum conservation and symmetrized with respect to $k = 0$ in Fig. 1g, h, Fig. 2a–d, and Supplementary Fig. 3. In the process of ARPES, we simultaneously recorded the photocurrent between the ground and samples (Fig. 2f) and green light emission by photoluminescence (Fig. 2g, inset). ARPES data of TiSe$_2$ (>99.995%, HQ graphene) in Fig. 2e were taken at Beamline I05, the Diamond Light Source.

### TEM experiments

HAADF-STEM and selected area electron diffraction (SAED) images have been collected with JEM-ARM200F equipped with a cold FEG and double Cs correctors (JEOL Ltd.) at 200 kV. A single-tilt cryo-transfer holder (Elsa 698, Gatan Inc.) was used to keep the PbI$_2$ samples at cryogenic temperature. PbI$_2$ flakes were mechanically exfoliated on the 300 nm SiO$_2$/Si substrate. To transfer PbI$_2$ flakes onto a Quantifoil holey carbon TEM grid (SPI Supplies Inc), we used the dry transfer method using isopropanol (IPA). The TEM grid was placed directly on the target flakes and a droplet of IPA was dropped on the TEM grid. After drying, the sample was kept in the N$_2$-filled glove box overnight to enhance adhesion between the carbon film of TEM grid and PbI$_2$. PbI$_2$/TEM grid was detached from substrates by dropping IPA. SAED patterns were obtained to identify the crystallographic orientation of the transferred PbI$_2$ flakes. After TEM measurements, the PbI$_2$/TEM grid was attached to SiO$_2$/Si substrate again with IPA and dried overnight to improve the adhesion between PbI$_2$ and SiO$_2$/Si substrate and TEM grid was removed by a tweezer without IPA. Focused ion beam (crossbeam 540, ZEISS) with a Ga source was used to fabricate cross-sectional TEM samples from the PbI$_2$ flakes re-transferred on the substrate. Crystallographic orientation information was used to choose the zone axis of the TEM samples.

### DFT calculations

All the DFT calculations have been carried out by using the Vienna Ab initio Simulation Package (VASP) code[47,48] employing the projector augmented wave (PAW) method[49,50]. The $5d^{10} 6s^2 6p^2$ and $5s^2 5p^5$ were explicitly considered as the valence electron configurations of Pb and I, respectively, within the PAW approach. The kinetic cutoff energy for the plane wave basis set was set to 500 eV, while the irreducible Brillouin zone integration was sampled by using the Γ-centred $k$-point meshes of $11 \times 11 \times 3$ and $11 \times 11 \times 6$ for the 4H and 2H phases, accordingly. The semi-local exchange-correlation ($xc$) functional due to Perdew, Burke, and Ernzehof (PBE)[51] was used to treat the DFT exchange-correlation energy using the generalized gradient approximation, while correcting for the weak van der Waals interactions via the Grimme's DFT-D2 scheme[52]. In this work, both the unit cell vectors and atomic positions were fully relaxed using a force and total energy tolerance of 0.01 eV Å$^{-1}$ and $10^{-5}$ eV, respectively. For the energy differences calculated by displacing the upper layer Pb atoms in 4H-PbI$_2$ (see Fig. 4c, d), a uniform grid along the cartesian $x$- and $y$-coordinates was generated by considering increments of 0.181 Å and 0.157 Å, respectively. Here, only the unit cell in the $z$-direction and the $z$-coordinates of the Pb atoms were relaxed while constraining the other two directions. The I atoms were fully relaxed. To account for a more accurate description of the band structure of PbI$_2$, the hybrid DFT $xc$ functional due to Heyd, Scuseria, and Ernzerhof (HSE06)[53,54] was used to calculate the electronic band structures of PbI$_2$. Specifically, the spin-orbit coupling effects were included here perturbatively, with a scissor operator correction[55,56] of 0.3 eV to match the experimental band gaps reported in the literature[28].

## Data availability

All data generated in this study are provided in the article and Supplementary Information, and raw data are provided in the Source Data file. Source data are provided with this paper. Additional data and materials are available from the corresponding authors upon request. Source data are provided with this paper.

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

## Acknowledgements

This work was supported by the National Research Foundation (NRF) of Korea funded by the Ministry of Science and ICT (grant number NRF-2021R1A3B1077156, NRF-2020K1A3A7A09080364, NRF-RS-2022-00143178 to K.S.K., NRF-2018M3D1A1058536 to A.S., NRF-2022R1A2C4002559 to K.K., and NRF-2017R1A5A1014862 to K.S.K. and K.K.), the Yonsei Signature Research Cluster Program funded by Yonsei University (2022-22-0004 to K.S.K. and K.K.), and Korea Institute for Advancement of Technology (KIAT) grant funded by Korea Government (MOTIE) (P0002019 to A.S., Human Resource Development Program for Industrial Innovation). This research used resources of the Advanced Light Source, which is the DOE Office of Science User Facility under the contract number DE-AC02-05CH11231. Computational resources have been kindly provided by the KISTI Supercomputing Center (KSC-2022-CRE-0038). The part of this work on 1T-TiSe2 was carried out with the support of the Diamond Light Source (beamline I05). XRD experiments were carried out using RIGAKU, SmartLab at the KAIST analysis center for research advancement (KARA).

## Author contributions

K.K., A.S., and K.S.K supervised the project. S.C. and S.H.R. performed ARPES experiments with help from Y.S., G.A., C.K., and M.K. S.L. performed TEM experiments. G.L. performed DFT calculations. S.C., G.L., S.L., K.K., A.S., and K.S.K wrote the manuscript with contributions from all other co-authors.

## Competing interests

The authors declare no competing interests.
