## [Peer Review File · Nature Communications]

Order-disorder phase transition driven by interlayer sliding in lead iodidesREVIEWER COMMENTS

Reviewer #1 (Remarks to the Author):

I read the manuscript "Order-disorder phase transition driven by interlayer sliding in lead iodides" by Cha et al. with great interest. In this work, the authors elegantly combine angle-resolved photoemission (ARPES) and electron microscopy techniques with density functional theory (DFT) to study the prototypical layered material PbI₂. This is one of the more interesting materials in which relatively stable I-Pb-I trilayers, hereafter referred to simply as layers, stack in many different ways to form different polytypes. In this respect, PbI₂ appears to be a particularly complex case, as a large number of different polytypes have been experimentally demonstrated over the past 60 years. However, despite this great research effort, it is still unclear which polytype is the ground state at low temperatures. This is indeed important since the optical and electronic properties of this technologically relevant material could change depending on the stacking order. In the present manuscript, the authors address this question in a somewhat indirect but elegant way: By comparing ARPES measurements with theoretical band structures determined by DFT, they conclude that the low-temperature phase of PbI₂ is not the 2H polytype (commonly and correctly referred to as the 1T structure in the context of transition metal dichalcogenides), but the 4H polytype. This finding is further supported by high-resolution STEM measurements. Furthermore, the authors show that the system undergoes an order-disorder transition with increasing temperature above 120 K, characterized by in-plane shifts of the layers. This transition is accompanied by significant changes in photocurrent and photoluminescence. Careful analysis of the data suggests that the layers shift with respect to each other along low energy paths, similar to a melting transition. This previously largely neglected mechanism is indeed of general interest, as similar effects can occur in a wide variety of layered materials.

The manuscript is clearly written and the presentation of the results is to the point. It should be easy to follow even for non-expert readers. I think the reported mechanism is very important as it probably plays a key role in many similar materials but hasn't received much attention from the community yet. Therefore, I think this work is suitable to be published in Nature Communications. However, I found a few caveats that should be addressed first:

i) My first point concerns the chosen methodology which is on the one hand very elegant but on the other hand also somewhat indirect. It is known that these layered materials and in particular their stacking orders are quite sensitive to external parameters like pressure or uni-axial strain. In the present study the single crystal of PbI₂ has been glued on a sample holder and cooled down to base temperature which unavoidably strains the sample. Also, the treatment for the STEM experiments may put the sample in a different state than this of a free-standing single crystal. Therefore, I think it is not entirely clear whether the observed 4H polytype is really the "only true" low-temperature state of PbI₂ or whether the experimental conditions have tuned the system to this state. The large discrepancy in the reported stacking structures of PbI₂ may also indicate that all of these polytypes are energetically very close, and minute differences in experimental conditions ultimately determine which one is observed. In my opinion, non-destructive structure-sensitive methods on nearly free-standing single crystals such as X-ray diffraction would be the best way to determine the "true" low-temperature structure. However, I realize that this is not the main point of this manuscript, but I suggest that the authors discuss these points in some detail.

ii) In line 149-154 the authors argue that the layer sliding occurs along the x-direction (-120-direction). To this end, they show in Fig. S4 that no sliding in the y-direction can be observed. However, later in the text, the DFT analysis shows that according to the threefold lattice symmetry, the three sliding directions (-120), (2-10) and (-1-10) are equivalent and therefore should occur in the material. If we now consider the crystal structure along the (-120)-direction as in Fig. S4 a) the sliding vectors (2-10) and (-1-10) would give finite displacements in the y-direction. This would then lead to stripes or additional peaks in the Fourier-transform-images shown in Fig S4 c-f) similar to Fig. 3f). However, such features do not appear. Thus, the question arises as to why only one sliding vector out of three symmetry-equivalent sliding vectors occurs? Perhaps the sample preparation process caused some anisotropy favoring one of the slip vectors? I think this aspect is not really clear in the current version of the manuscript and should be discussed in more detail.

iii) In Fig. 2 f,g), the authors nicely show how the transition from order to disorder affects the photocurrent and photoluminescence. However, they hardly address how these changes can be reconciled with the observed changes in the electronic structure. I would like to see this aspect discussed in a bit more detail.

Best,
Tobias

Reviewer #2 (Remarks to the Author):

Seyeong Cha, et al., report the observation of an order-disorder phase transition in PbI₂. I found the manuscript is interesting. More and more researchers in the community of 2D materials begin to recognize the importance of interlayer interaction. The authors found a clear phase transition driven by interlayer interaction in PbI₂ and show how this phase transition affects the optoelectronic and optical characteristics of this material. However, I cannot recommend the publication of this manuscript in its present form. My concerns are the followings.

1. I found it is hard for me to understand the correlation between the change of lattice and the change of sample properties such as electronic structure, photocurrent, and etc. Do the authors have an explanation why the interlayer sliding of Pb could result in the broadening of APRES spectra? How does this so-called order-disorder phase transition result in a change of optoelectronic characteristic of PbI₂ as observed by the authors in Fig. 2g?

2. The disorder or the randomness of interlayer sliding has not been well proved by the authors in the manuscript. Can this phase transition at 120 K be detected in bulk materials by the bulk-sensitive methods like transport measurements or X-ray diffraction? If so, the randomness of interlayer sliding could be directly observed by the diffraction experiments. The authors may also consider show more STEM images taken in a larger field of vision that can cover more PbI₂ layers.

Reviewer #3 (Remarks to the Author):

In this paper Cha and colleagues report transmission electron microscopy (TEM), angle-resolved photoemission spectroscopy (ARPES) and other measurements on two-dimensional layered PbI₂.

They claim that a correction is needed to the commonly held idea that the low temperature structure of PbI₂ is the 2H polytype. Next, they observe a transition at 120 K that is attributed to an order-disorder transition. The existence of an apparent phase transition is supported by ARPES, photocurrent, and photoluminescence observations. It is reasonably attributed to a transition towards a disorder in the interlayer registry, which the Authors describe as distinct from stacking disorder because the layer-layer displacement can take any one of a continuum of in-plane vectors rather than a few discrete ones ('sliding disorder'). This is supported via TEM observations and further elucidated using density-functional theory calculations.

The paper is well-written, and I appreciate the clarity of the explanations. I think in principle the findings and associated discussion are important and deserve the extra reach afforded by publication in Nature Communications. However, below I point out a couple of issues which I hope the Authors will address before I would recommend the paper to be accepted for publication.

The first of the Authors' major claims is that the low-temperature structure of PbI₂ is not 2H, but rather 4H. I am confused by the Authors' diagrams in Fig. 1(b) and (c) that are supposed to show the 2H and 4H polytypes. I think that what is shown in Fig. 1(b) should actually be called the 1T polytype, because only one PbI₂ molecular unit is contained in the primitive cell as shown (and it

has octahedral coordination). And what is shown in Fig. 1(c) should be called 2H because there are two molecular units in the primitive cell. I would imagine that 4H-PbI₂ would appear as depicted in Fig. 2(a) of Journal of Materials Science 55, 10656–10667 (2020), for example.

As a related issue, I am guessing that the TEM image in Fig. 1(d) is actually consistent with the 2H polytype.

Given the above points, are the Authors sure that the calculated band structure shown in Figs. 1(e) and (f) (and in the SI) are for the 1H and 2H polytypes as depicted in (b) and (c), or for the 2H and 4H polytypes?

Perhaps I am wrong, but if not, I think the Authors need to clarify these issues before the paper can proceed towards publication. However, I understand that these issues do not have a significant impact on the second major claim of the paper, namely, the order-disorder transition at $T \sim 120$ K, that I think is very clearly supported by the Authors' observations and arguments.

But here I have another query which I would also like to see the Authors answer, related to the broadening of energy distribution curves (EDCs) as a function of temperature:

I understand that the introduction of disorder above $T = 120$ K will lead to a much shorter out-of-plane coherence length that should cause the intrinsic width of the valence bands to increase. (In the Supplementary Information, the Authors show band structures with significant out-of-plane dispersion for the relevant bands.) My question is about the expected effect of temperature in absence of sliding disorder (e.g., for TiSe₂, or for the low-temperature regime for PbI₂). In Fig. 2(a), the FWHM inferred from the EDC in the low-temperature region is compared against $k_B T$ [The $k_B T$ line doesn't extrapolate down through zero, and I assume this is because it is being added to some intrinsic ($T = 0$ K) width.] I would like to see an explicit explanation of the mechanism causing $k_B T$ broadening of the FWHM for the valence band EDCs. Exactly why does this happen? In other recent temperature dependent ARPES observations [for example, Nature Communications 11, 4215 (2020)], it does not seem either necessary or appropriate to account for thermal broadening in this way. Also, exactly why does the width continue to increase linearly, and more rapidly, within the disordered phase?

Other than the above issues, I find the Authors' results and arguments convincing and I do not have further questions.

Response Letter for NCOMMS-22-46824

We sincerely thank reviewers for your time and efforts in reviewing our manuscript. We are pleased to find that all the 3 reviewers commonly acknowledged the novelty, general interest, potential impact of our findings, and the clarity and readability of our manuscript. The reviewers raised a few points to be addressed in revisions, which were extremely helpful in improving our manuscript. Considering all the reviewers' points in full, we have carefully made revisions to the manuscript, as explained in detail below. We hope that all the reviewers will find our revisions satisfactory.

Point-by-point response to Reviewer #1:

I read the manuscript "Order-disorder phase transition driven by interlayer sliding in lead iodides" by Cha et al. with great interest. In this work, the authors elegantly combine angle-resolved photoemission (ARPES) and electron microscopy techniques with density functional theory (DFT) to study the prototypical layered material Pbl₂. This is one of the more interesting materials in which relatively stable I-Pb-I trilayers, hereafter referred to simply as layers, stack in many different ways to form different polytypes. In this respect, Pbl₂ appears to be a particularly complex case, as a large number of different polytypes have been experimentally demonstrated over the past 60 years. However, despite this great research effort, it is still unclear which polytype is the ground state at low temperatures. This is indeed important since the optical and electronic properties of this technologically relevant material could change depending on the stacking order. In the present manuscript, the authors address this question in a somewhat indirect but elegant way: By comparing ARPES measurements with theoretical band structures determined by DFT, they conclude that the low-temperature phase of Pbl₂ is not the 2H polytype (commonly and correctly referred to as the 1T structure in the context of transition metal dichalcogenides), but the 4H polytype. This finding is further supported by high-resolution STEM measurements. Furthermore, the authors show that the system undergoes an order-disorder transition with increasing temperature above 120 K, characterized by in-plane shifts of the layers. This transition is accompanied by significant changes in photocurrent and photoluminescence. Careful analysis of the data suggests that the layers shift with respect to each other along low energy paths, similar to a melting transition. This previously largely neglected mechanism is indeed of general interest, as similar effects can occur in a wide variety of layered materials.

The manuscript is clearly written and the presentation of the results is to the point. It should be easy to follow even for non-expert readers. I think the reported mechanism is

very important as it probably plays a key role in many similar materials but hasn't received much attention from the community yet. Therefore, I think this work is suitable to be published in Nature Communications.

Our reply: We thank the reviewer for highly positive assessments and careful reading of our manuscript, which is very well summarized above. We feel that the reviewer saw though every point that we thought of as the strength of our work. The reviewer also raised a few points as below, which we find very constructive and extremely helpful in improving our manuscript. We have carefully addressed all the reviewer's points in full, and hope that you will find our revisions satisfactory.

However, I found a few caveats that should be addressed first:

i) My first point concerns the chosen methodology which is on the one hand very elegant but on the other hand also somewhat indirect. It is known that these layered materials and in particular their stacking orders are quite sensitive to external parameters like pressure or uni-axial strain. In the present study the single crystal of Pbl₂ has been glued on a sample holder and cooled down to base temperature which unavoidably strains the sample. Also, the treatment for the STEM experiments may put the sample in a different state than this of a free-standing single crystal. Therefore, I think it is not entirely clear whether the observed 4H polytype is really the "only true" low-temperature state of Pbl₂ or whether the experimental conditions have tuned the system to this state. The large discrepancy in the reported stacking structures of Pbl₂ may also indicate that all of these polytypes are energetically very close, and minute differences in experimental conditions ultimately determine which one is observed. In my opinion, non-destructive structure-sensitive methods on nearly free-standing single crystals such as X-ray diffraction would be the best way to determine the "true" low-temperature structure. However, I realize that this is not the main point of this manuscript, but I suggest that the authors discuss these points in some detail.

Our reply: Thank you for constructive and helpful comments. We couldn't agree more with the reviewer's interesting expression "indirect but elegant". As a group in the field of ARPES, we believe that it is not only possible, but also important to trace back the real-space crystal structure from some "fingerprints" left in high-resolution ARPES data. Not always but sometimes, this approach makes sense in that even a very subtle real-space structural variation, which is difficult to directly identify with diffraction methods, may be manifested as sizable changes in the momentum-space electronic structure that can be more easily identified by high-resolution ARPES. In this sense, the number of valence bands in Fig. 1g and apparent periodicity in Fig. 1i clearly demonstrate that the low-temperature phase is at least not 2H as known before. If you also consider the

real-space image shown in Fig. 1d, taken by another experimental method (STEM), the low-temperature phase can be specifically identified as the 4H polytype.

Since ARPES and STEM are two independent experimental methods, it leaves a very small room that the two methods by chance indicate the same structure. Nevertheless, we agree on the reviewer' points that (1) there are many energetically close polytypes in PbI_2 , and that (2) this stacking order may be sensitive to external parameters. In fact, less attention was paid to the possibility of employing a non-destructive XRD method, because some of recent works already reported that there exist peaks expected for the 4H phase [Chem. Mater. **18**, 2059, (2016); Ref. 32, J. Mater. Sci. **51**, 9123 (2016); Ref. 33]. To make things perfectly clear, we performed XRD measurements on our PbI_2 samples at low temperatures, as shown in Fig. R1 below. We found the clear signature of (201), (203), and (103) peaks indicated by red arrows, which are expected for the 4H polytype (ICDD card no. 04-007-3144) and fully consistent with previous reports (Refs. 32,33).

In the revised manuscript, we included XRD data in Fig. R1 as Supplementary Fig. 2, and clearly explained that the low temperature 4H structure can be supported even by non-destructive XRD experiments.

Figure R1 | XRD measurements. XRD data taken from our single-crystal PbI_2 samples at 100 K and plotted as a function of 2θ . Red arrows indicate the peaks expected for the 4H phase (ICDD card no. 04-007-3144). This is consistent with previous XRD reports that exhibited the similar 4H peaks (Refs. 32,33).

ii) In line 149-154 the authors argue that the layer sliding occurs along the x-direction (-120-direction). To this end, they show in Fig. S4 that no sliding in the y-direction can be

observed. However, later in the text, the DFT analysis shows that according to the threefold lattice symmetry, the three sliding directions (-120) , $(2-10)$ and $(-1-10)$ are equivalent and therefore should occur in the material. If we now consider the crystal structure along the (-120) -direction as in Fig. S4 a) the sliding vectors $(2-10)$ and $(-1-10)$ would give finite displacements in the y -direction. This would then lead to stripes or additional peaks in the Fourier-transform-images shown in Fig S4 c-f) similar to Fig. 3f). However, such features do not appear. Thus, the question arises as to why only one sliding vector out of three symmetry-equivalent sliding vectors occurs? Perhaps the sample preparation process caused some anisotropy favoring one of the slip vectors? I think this aspect is not really clear in the current version of the manuscript and should be discussed in more detail.

Our reply: We thank the reviewer for this helpful point. It is true that the layer sliding seems to occur along one of the three equivalent directions in Supplementary Fig. 4b. However, it should be considered that this image is taken over the relatively small area of $3 \times 4.5 \text{ nm}^2$, in which one of the three shifts can be dominant. Indeed, SAED patterns taken in y over the wider area of $80 \times 80 \text{ nm}^2$ show abrupt broadening of peaks near T_c (Fig. R2, below) towards the formation of streaks, as exactly expected by the reviewer. Note, however, that in y or $\langle 100 \rangle$ directions the spots cannot be as streaky as those in x or $\langle 110 \rangle$ directions (Fig. 3e, f), because this is about 30° off the sliding directions.

Figure R2 | Wide-scale SAED and temperature dependence. **a** SAED patterns of PbI_2 at 100 K, taken in the y direction over the wide region of $80 \times 80 \text{ nm}^2$. **b** FWHM of SAED peaks plotted as a function of temperature. This data demonstrates that the signature of interlayer sliding above T_c can also be captured in SAED patterns taken over the area of $80 \times 80 \text{ nm}^2$ towards the formation of streaks as observed in Fig. 3e, f.

This kind of triple-domain separations is not something unique but has been widely observed in intermediate phases between the ordered and the fully disordered phases [e. g., Commun. Phys. **4**, 229 (2021)]. This is indicative of remnant interlayer coupling, which cannot be captured in DFT considering bilayer PbI_2 . Without these explanations,

however, we understand that Supplementary Fig. 4b in the previous manuscript may be confusing, as pointed out by the reviewer. In the revised manuscript, we explained the relative scale of STEM data and the signature of interlayer sliding in the y direction with Fig. R2 shown above (this is now part of Supplementary Fig. 6).

iii) In Fig. 2 f,g), the authors nicely show how the transition from order to disorder affects the photocurrent and photoluminescence. However, they hardly address how these changes can be reconciled with the observed changes in the electronic structure. I would like to see this aspect discussed in a bit more detail.

Our reply: We are glad that the reviewer brought up this point. As the reviewer might expect, at this stage we were conservative in making any conclusive arguments on the connection between electronic structures and transport and optoelectronic properties, one of the long-term goals in the field of condensed-matter physics. As encouraged by the reviewer, however, we elaborated the picture we have based on our new findings:

As for the photocurrent, we found a clear dip at 160 K, which is a peak in resistivity. The similar anomalous peak in resistivity near T_c was found in other phase transitions [for example, see Fig. 2(c), Phys. Rev. B **99**, 195142 (2019)]. This kind of peaky anomaly was attributed to either a crossover in the type of carriers or divergent scattering in the vicinity of T_c due to the coexistence of competing phases and their phase fluctuations [see Phys. Rev. Lett. **31**, 241 (1973)]. For the latter, the effect of inhomogeneity near T_c makes charge carriers strongly localized, which explains the peak in resistivity. Indeed, this peak in resistivity is also manifested by charging effects in Pb $5d$ core-level spectra (Supplementary Fig. 4).

As for photoluminescence, we found an abrupt change in the slope of PL intensity versus temperature. The similar anomalous temperature dependence of PL associated with the order-disorder phase transition was found in other systems [for example, see Fig. 4(b), Adv. Mater. **30**, 1705801 (2018)]. In the disorder phase, the energy and crystal momenta of conduction and valence bands (and so the bandgap) are ill-defined with considerable uncertainty of over $50k_B T$. Then, the scattering of electrons and holes is dominated by various sources of disorder potentials rather than the radiative electron-hole re-combinations. In this sense, the change in the slope of PL seems to suggest the crossover between two regimes, where quasiparticle scattering is dominated by either intrinsic (phonon and electron-hole recombination) or extrinsic (disorder) factors.

Best, Tobias

Our reply: Tobias!! Thank you so much for this constructive and helpful review report, and I really appreciate it!

Point-by-point response to Reviewer #2:

Seyoung Cha, et al., report the observation of an order-disorder phase transition in Pbl₂. I found the manuscript is interesting. More and more researchers in the community of 2D materials begin to recognize the importance of interlayer interaction. The authors found a clear phase transition driven by interlayer interaction in Pbl₂ and show how this phase transition affects the optoelectronic and optical characteristics of this material. However, I cannot recommend the publication of this manuscript in its present form. My concerns are the followings.

Our reply: We thank the reviewer for time and efforts in reviewing our manuscript. We also appreciate the reviewer's highly positive assessments on the interest, importance, and clarity of our findings. The reviewer requested us to address two concerns below, which we find helpful in improving our manuscript. Considering these two concerns, we have carefully made appropriate changes to the manuscript as summarized below, and hope that the reviewer will find our revisions satisfactory.

1. I found it is hard for me to understand the correlation between the change of lattice and the change of sample properties such as electronic structure, photocurrent, and etc. Do the authors have an explanation why the interlayer sliding of Pb could result in the broadening of APRES spectra? How does this so-called order-disorder phase transition result in a change of optoelectronic characteristic of Pbl₂ as observed by the authors in Fig. 2g?

Our reply: We thank the reviewer for this question. Let us first clarify the correlation between lattice-structure and electronic-structure changes. Once the interlayer sliding occurs, there is no long-range order in the out-of-plane direction with spatially varying (disordered) interlayer hopping. It follows that there is no singly defined Brillouin zone and electronic structure, and ARPES probes only coherent part of electronic structures with the spread of energy and momenta. In other words, this spectral broadening can be more intuitively described by the superposition of electronic-structure continua.

Let us now explain why the order-disorder phase transition results in the change of optoelectronic characteristics. Once the phase transition occurs, the energy and crystal momenta of conduction and valence bands (and so the bandgap) are ill-defined with considerable uncertainty of over $50k_B T$. Then, the scattering of electrons and holes is dominated by various sources of disorder potential rather than the radiative electron-hole re-combinations. In this sense, the change in the slope of PL seems to suggest the crossover between two regimes, where quasiparticle scattering is dominated by either intrinsic (phonon and electron-hole recombination) or extrinsic (disorder) factors.

In the revised manuscript, we provide clear explanations of the correlation between lattice-structure and electronic-structure changes and of that between order-disorder phase transitions and optoelectronic characteristics.

2. The disorder or the randomness of interlayer sliding has not been well proved by the authors in the manuscript. Can this phase transition at 120 K be detected in bulk materials by the bulk-sensitive methods like transport measurements or X-ray diffraction? If so, the randomness of interlayer sliding could be directly observed by the diffraction experiments. The authors may also consider show more STEM images taken in a larger field of vision that can cover more Pbl₂ layers.

Our reply: We thank the reviewer for this constructive and helpful point. We agree on the point that, although we demonstrated the presence of disorder by the elongation of peaks (Fig. 3f) taken by the Fourier transform of relatively small-area STEM images in Fig. 3b, we have not demonstrated the randomness of interlayer sliding with the wider area diffraction experiments. Motivated by the reviewer's comment, we present below the diffraction pattern taken for the wide scale of $80 \times 80 \text{ nm}^2$. It clearly shows streaks indicated by red arrows, which is conclusive evidence for the randomness of interlayer sliding. In the revised manuscript, we included Fig. R3 below as Supplementary Fig. 5.

Figure R3 | Diffraction pattern taken in the larger field of view. **a** SAED patterns of Pbl₂ taken for the larger area of $80 \times 80 \text{ nm}^2$ at 300 K along the x direction. **b,c** Part of SAED patterns indicated by the yellow dotted box in **a**, taken at **(b)** 300 K and **(c)** 100 K. The yellow ovals indicate the spotty 4H peaks, and the red arrows show the formation of streaks, which is clear evidence for the randomness of interlayer sliding.

Point-by-point response to Reviewer #3:

In this paper Cha and colleagues report transmission electron microscopy (TEM), angle-resolved photoemission spectroscopy (ARPES) and other measurements on two-dimensional layered Pbl₂.

They claim that a correction is needed to the commonly held idea that the low temperature structure of Pbl₂ is the 2H polytype. Next, they observe a transition at 120 K that is attributed to an order-disorder transition. The existence of an apparent phase transition is supported by ARPES, photocurrent, and photoluminescence observations. It is reasonably attributed to a transition towards a disorder in the interlayer registry, which the Authors describe as distinct from stacking disorder because the layer-layer displacement can take any one of a continuum of in-plane vectors rather than a few discrete ones ('sliding disorder'). This is supported via TEM observations and further elucidated using density-functional theory calculations.

*The paper is well-written, and I appreciate the clarity of the explanations. I think in principle the findings and associated discussion are important and deserve the extra reach afforded by publication in *Nature Communications*. However, below I point out a couple of issues which I hope the Authors will address before I would recommend the paper to be accepted for publication.*

Our reply: We thank the reviewer for highly positive assessments on the importance of our findings and the clarity of explanations. We are pleased to find that the reviewer in principle recommended publication in *Nature Communications*, but before doing so, the reviewer raised a few points to be addressed. Considering all the reviewer's points, we have carefully made appropriate revisions, as summarized in detail below.

*The first of the Authors' major claims is that the low-temperature structure of Pbl₂ is not 2H, but rather 4H. I am confused by the Authors' diagrams in Fig. 1(b) and (c) that are supposed to show the 2H and 4H polytypes. I think that what is shown in Fig. 1(b) should actually be called the 1T polytype, because only one Pbl₂ molecular unit is contained in the primitive cell as shown (and it has octahedral coordination). And what is shown in Fig. 1(c) should be called 2H because there are two molecular units in the primitive cell. I would imagine that 4H-Pbl₂ would appear as depicted in Fig. 2(a) of *Journal of Materials Science* 55, 10656–10667 (2020), for example.*

As a related issue, I am guessing that the TEM image in Fig. 1(d) is actually consistent with the 2H polytype. Given the above points, are the Authors sure that the calculated band

structure shown in Figs. 1(e) and (f) (and in the SI) are for the 1H and 2H polytypes as depicted in (b) and (c), or for the 2H and 4H polytypes? Perhaps I am wrong, but if not, I think the Authors need to clarify these issues before the paper can proceed towards publication. However, I understand that these issues do not have a significant impact on the second major claim of the paper, namely, the order-disorder transition at $T \sim 120$ K, that I think is very clearly supported by the Authors' observations and arguments.

Our reply: Thank you so much for helpful comments. Let us clarify that this confusion comes from the matter of nomenclature. The reviewer is correct that what is shown in Fig. 1b is typically termed "1T" in the nomenclature for the polytype of TMDs (MX_2), where "1" means the number of X-M-X layers in the unit cell, and "T" means that the type of crystal systems is trigonal. However, the polytype of PbI_2 (and silicon carbides) has been conventionally described by a different nomenclature [*Cryst. Res. Technol.* **45**, 455 (2010)], where "2" means the number of iodine layers in the unit cell, and "H" means that the type of lattice systems is hexagonal, as summarized in Fig. R4 below.

Ramsdell notation: nY	n	Y
Nomenclature for TMDs (MX_2)	Number of X-M-X layers	Type of crystal systems
Nomenclature for PbI_2	Number of iodine layers	Type of lattice systems

Figure R4 | Summary of different nomenclatures between TMDs and PbI_2 .

As for what is shown in Fig. 1c, the number of iodine layers in the unit cell is 4, and the type of lattice systems is hexagonal. Hence, it is called "4H" in the nomenclature of PbI_2 . Although we have known the fact that the similar structure with Fig. 1c is called "2H" in the nomenclature of TMDs, we chose to use the nomenclature of PbI_2 , because most of publications in literature on PbI_2 have conventionally used that nomenclature, except for a few examples. One of such examples is in fact mentioned by the reviewer,

Figure 2a of J. Mater. Sci. **55**, 10656 (2020). In this paper, PbI_2 is compared with MoSe_2 , which is why the nomenclature of TMDs was applied to PbI_2 as well.

Thanks to the reviewer's comments, however, we realized that this nomenclature of PbI_2 is very confusing especially for those who are familiar with that of TMD materials. To prevent any confusion, we made the following revisions to the revised manuscript: (1) we explicitly mention that the nomenclature of PbI_2 is different from that of TMDs at the first time introduced, and (2) the nomenclature of TMDs is written in parenthesis together with that of PbI_2 at several important points, such as the introduction and the legends of Fig. 1 and Supplementary Fig. 1, and when we explain Fig. 1b and Fig. 1c.

But here I have another query which I would also like to see the Authors answer, related to the broadening of energy distribution curves (EDCs) as a function of temperature: I understand that the introduction of disorder above $T=120$ K will lead to a much shorter out-of-plane coherence length that should cause the intrinsic width of the valence bands to increase. (In the Supplementary Information, the Authors show band structures with significant out-of-plane dispersion for the relevant bands.) My question is about the expected effect of temperature in absence of sliding disorder (e.g., for TiSe_2 , or for the low-temperature regime for PbI_2). In Fig. 2(a), the FWHM inferred from the EDC in the low-temperature region is compared against $k_B T$ [The $k_B T$ line doesn't extrapolate down through zero, and I assume this is because it is being added to some intrinsic ($T=0$ K) width.] I would like to see an explicit explanation of the mechanism causing $k_B T$ broadening of the FWHM for the valence band EDCs. Exactly why does this happen? In other recent temperature dependent ARPES observations [for example, Nature Communications 11, 4215 (2020)], it does not seem either necessary or appropriate to account for thermal broadening in this way. Also, exactly why does the width continue to increase linearly, and more rapidly, within the disordered phase?

Our reply: We are glad that the reviewer brought up this point. In fact, the $k_B T$ effect to ARPES spectral widths has been well-established by the model system of $\text{Mo}(110)$ [see T. Valla *et al.*, Phys. Rev. Lett. **83**, 2085 (1999); Ref. 39]. The EDC width of ARPES spectra is related to the lifetime of photohole (or the imaginary part of self-energy that reflects manybody interactions). This lifetime of photohole can be approximately described by mainly 3 decay mechanisms: electron-phonon scattering, electron-electron scattering, and electron-impurity scattering. Since electron-impurity scattering does not depend on the temperature unless there is an order-disorder phase transition, it constitutes a certain offset in the plot of EDC width as a function of temperature, which is the origin of "intrinsic width" mentioned by the reviewer. What depends on the temperature is then two-fold: electron-phonon scattering and electron-electron scattering. However, the temperature dependence of electron-electron scattering is typically very weak in

many materials, which leaves only electron-phonon scattering as the major factor that accounts for the temperature dependence of ARPES spectral widths.

As shown in the previous works [for example, see Fig. 2(b) and Fig. 4(a) in Ref. 39], the temperature dependence of ARPES widths driven by electron-phonon coupling is approximately linear with the slope of $2\pi\lambda k_B$, where λ is the electron-phonon coupling constant. Considering that λ is typically in the range of 0.1 ~ 0.5, one would expect the roughly $k_B T$ dependence of ARPES EDC widths. Of course, this slope may be different in other materials because the slope depends on λ , the characteristic of given materials.

When the order-disorder phase transition sets in, electron-impurity scattering is no longer constant with temperature. In this disordered phase, the slope of ARPES widths versus the temperature reflects how rapidly the source of electron-impurity scattering (or the degree of disorder) increases with temperature, which can be as large as $50k_B T$. This point is now more clearly explained with Ref. 39 in the revised manuscript.

Other than the above issues, I find the Authors' results and arguments convincing and I do not have further questions.

Our reply: Once again, we are grateful to the reviewer for highly positive assessments and hope you will find our revisions satisfactory.

REVIEWERS' COMMENTS

Reviewer #1 (Remarks to the Author):

The authors have thoroughly responded to the points raised by the reviewers and have revised their manuscript accordingly. They also provide additional XRD data to further support their findings. I stand by my earlier assessment and believe that this is a very interesting paper on this topic suitable for publication in Nature Communications.

All the best,
Tobias

Reviewer #2 (Remarks to the Author):

The authors addressed all my concerns in the revised manuscript. I recommend its publication in Nature Communications.

Reviewer #3 (Remarks to the Author):

I have read the Authors' responses and the corresponding revisions. The Authors have expertly answered my questions and, to the best of my understanding, those of the other reviewers. I am satisfied by their responses and indeed I learned a lot from them. I happily recommend this interesting and important paper to proceed to acceptance for publication in Nature Communications.